# Laboratory Cross-Sensitivity Evaluation of Low-Cost Electrochemical Formaldehyde Sensors

**DOI:** 10.3390/s25103096

**Published:** 2025-05-14

**Authors:** Zheyuan Pei, Kerry E. Kelly

**Affiliations:** Department of Chemical Engineering, University of Utah, Salt Lake City, UT 84112, USA; zheyuan.pei@utah.edu

**Keywords:** formaldehyde, low-cost electrochemical sensor, broadband cavity-enhanced absorption spectroscopy, cross-sensitivity

## Abstract

Formaldehyde is the most abundant carbonyl globally and the biggest driver of cancer risk in the United States among hazardous air pollutants. Ambient formaldehyde concentration measurements are generally sparse due to high measurement costs and limited measurement infrastructure. Recent studies have used low-cost air quality sensors to affordably improve spatial coverage and provide real-time measurements. Our previous research evaluated the laboratory performance of a low-cost electrochemical formaldehyde sensor (Sensirion SFA30) over formaldehyde concentrations ranging from 0 to 76 ppb. The sensors exhibited good linearity of response, a low limit of detection, and good accuracy in detecting formaldehyde. This study evaluated the cross-sensitivity of the SFA30 and the Gravity sensors (electrochemical formaldehyde sensors) over formaldehyde concentrations ranging from 0 to 326 ppb in a laboratory evaluation system, with broadband cavity-enhanced absorption spectroscopy used to obtain the reference measurements. We evaluated the sensors in a mixture of formaldehyde with five outdoor trace gases (CO, NO, NO_2_, O_3_, and isobutylene) and two indoor VOCs (methanol and isopropyl alcohol). The results suggest that the Gravity sensors may be useful for outdoor formaldehyde measurements when formaldehyde levels are well above background levels and that the SFA30 sensors may be useful screening tools for indoor environments, if properly calibrated.

## 1. Introduction

Formaldehyde is an important indoor and outdoor air pollutant due to its significant health and environmental impacts [1,2,3]. Formaldehyde is a known human carcinogen [4]. It is the most important air-pollutant driver of cancer risk in the U.S., with an estimated 84–160 people per year developing cancer due to exposure to ambient formaldehyde [4,5]. Formaldehyde also plays a key role in atmospheric chemistry and in ground-level pollutant formation. For example, photochemical oxidation of formaldehyde generates radicals that contribute to ozone and secondary organic aerosol formation, both of which are linked to poor air quality and adverse health effects [1,6,7]. Although over 25 million people are exposed to formaldehyde levels that exceed the 1 in 10^6^ cancer risk threshold [4], formaldehyde concentration measurements are generally sparse due to high measurement costs and limited measurement infrastructure [8]. The U.S. EPA’s Photochemical Assessment Monitoring Stations (PAMS) network utilizes spectroscopic methods for hourly averaged measurements of formaldehyde at 15 of its 43 monitoring sites [9]. However, ambient formaldehyde monitoring outside the PAMS network remains limited. Improving the temporal and geospatial distribution of formaldehyde measurements is critical for source attribution, emission regulation, and public health protection.

Current measurement techniques for formaldehyde, such as satellite-based, spectrometric, and optical techniques, are either expensive or labor-intensive [10]. Satellite-based measurements provide large-scale estimates of ambient formaldehyde concentrations but have a limited resolution (on the 10 km^2^ scale) [8,11]. Spectrometric techniques, including gas chromatography (GC) and high-performance liquid chromatography (HPLC), offer high accuracy and sensitivity but are often costly and complex to implement [12,13]. Optical methods, such as differential optical absorption spectroscopy (DOAS), are used for remote sensing and in situ monitoring, delivering real-time measurements, but are expensive and require specialized equipment [14,15]. In recent years, numerous studies have highlighted the use of low-cost sensors for air quality monitoring [16,17,18,19,20,21,22,23]. These more affordable sensors can provide greater spatial representation and real-time monitoring, although they may have limitations in terms of accuracy compared to traditional techniques.

Previous studies have reported laboratory evaluations of low-cost formaldehyde sensors (Table 1). However, most of these studies focused on formaldehyde concentration ranges that were higher than typical indoor (~17 ppb) [24] or ambient (~3 ppb) [25] levels in the United States. Many studies have evaluated metal oxide sensors, which require high formaldehyde concentrations (ppm level) and high operating temperatures. Few studies have reported laboratory evaluations of electrochemical sensors. In addition, limited studies have provided comprehensive performance metrics, such as linearity, sensor accuracy, precision, limit of detection (LOD), and cross-sensitivity.

It is critical to understand sensor cross-sensitivity to common outdoor and indoor trace gases before low-cost sensors are used for field measurements of formaldehyde. Crispi and Neri reported that their Al, Ca-doped ZnO sensor exhibited about twice the response to NO_2_ (5 ppm), about one-fifth of the response to NO (125 ppm), and a negligible response to CO (25 ppm) compared to formaldehyde (4 ppm) [31]. Pei et al. presented a preliminary cross-sensitivity evaluation of the electrochemical Sensirion SFA30 sensors to CO (39.7 ppm), NO (101 ppb), NO_2_ (83 ppb), and isobutylene (100 ppb) [10]. Although their study demonstrated that the Sensirion SFA30 sensors were sensitive to CO, NO, and NO_2_, a more comprehensive understanding of the sensors’ performance with a range of atmospheric trace gases is needed. Gautam et al. reported some cross-sensitivity of their MOS/SiNW nanocomposite sensor to methanol, ethanol, and acetone at room temperature, with each of these gases (1 ppm) causing approximately 50% of the sensor response compared to formaldehyde only [28]. Li et al. found that their hollow TiO_2_ microsphere sensor exhibited a ten-fold higher response to formaldehyde compared to other gases like methylbenzene, methanol, ethanol, acetone, and ammonia (all gases of 3 ppm, at 20 °C and 40% RH) [34]. While these studies revealed some cross-sensitivity of the low-cost formaldehyde sensors, they tended to lack a quantitative characterization of sensor cross-sensitivity. Consequently, a thorough understanding of the cross-sensitivity of electrochemical formaldehyde sensors is needed.

This study builds on a previous study [10] and evaluates the laboratory performance of two electrochemical sensors (Sensirion SFA30^®^, Stäfa, Switzerland and DFRobot Gravity^®^, Shanghai, China) over the formaldehyde concentration range of 11.8–255 ppb compared to high-accuracy broadband cavity-enhanced absorption spectroscopy (BBCEAS) as the reference instrument [37]. It also systematically evaluates the cross-sensitivity of the sensors to common outdoor trace gases (CO, NO, NO_2_, O_3_, and isobutylene, isobutylene being an example of an ambient VOC) and two potential indoor VOCs, methanol and isopropyl alcohol (IPA). This study will help determine the suitability of the sensors for applications in indoor or outdoor environments.

## 2. Materials and Methods

This study evaluated two types of electrochemical formaldehyde sensors: the Sensirion SFA30 (four) and the DFRobot Gravity (two) (Appendix A and Table 2). We had originally planned to evaluate three Gravity sensors, but one sensor malfunctioned during the evaluation. We also attempted to evaluate a third formaldehyde sensor (SGX PS1-HCHO-1, Neuchatel, Switzerland). However, the SGX PS1-HCHO-1 provided no signal when exposed to formaldehyde at concentrations as high as 60 ppb, so this sensor was not further evaluated.

### 2.1. Laboratory Evaluation System

The evaluation entailed exposing the low-cost sensors to target concentrations of formaldehyde and interference gases. Each sensor was placed in a custom-made 3D-printed polylactic acid (PLA) chamber, which was sealed with polytetrafluoroethylene (PTFE) to minimize leakage (Appendix A). The four SFA30 sensors were connected to one Arduino Nano board via a multiplexer (I^2^C protocol), and the three Gravity sensors were connected to three Arduino Uno boards (UART protocol). Both types of sensors reported measurements at sampling intervals of 10 s. The Arduino boards were connected to a computer, which allowed real-time viewing and recording of the measurements.

Figure 1 illustrates the laboratory system used to evaluate the low-cost formaldehyde sensors. The system included a broadband cavity-enhanced absorption spectrometer (BBCEAS) as the reference formaldehyde measurement device [10]. The system contained three gas lines, a blank line, a sample line, and an RH line. The blank line contained zero air (generated by a Teledyne T701, San Diego, CA, USA), which provided the baseline signal for both the sensors and the BBCEAS. The sample line provided a mixture of formaldehyde and interference gas. Formaldehyde was generated in a permeation oven (Dynacalibrator^®^ Model 340, VICI, Inc., Poulsbo, WA, USA) by heating a permeation tube of paraformaldehyde (325 ng/min; VICI, Inc., Poulsbo, WA, USA) to 90 °C. Interference gas was mixed into the sample line using a stainless-steel tee connector. The interference gas tests were categorized as outdoor trace gas tests (CO, NO, NO_2_, O_3_, and isobutylene) and indoor VOC tests (methanol and IPA). Each interference gas (CO, NO, NO_2_, isobutylene, methanol, and IPA) was obtained by diluting a calibration gas cylinder with zero air. O_3_ was generated by the Model 306 Ozone Calibration Source (2B Technologies, Broomfield, CO, USA). We also used reference/research-grade instruments after the BBCEAS to obtain the concentration of each interference gas. Section 2.4 provides details of the reference/research-grade instruments.

Both the blank line and the sample line were set to a flow rate of 1 LPM using a mass flow controller (MFC, Cole Parmer, Inc., Vernon Hills, IL, USA). A switching valve (XM40NBG45A, Parker, Inc., Mineral Wells, TX, USA) allowed the two lines to alternate. The RH line provided essential relative humidity (RH) to the electrochemical sensors by flowing 0.15/0.35 LPM (outdoor trace gas tests/indoor VOC tests) zero air through a 6″ Nafion (OD 0.110″, Perma Pure LLC, Lakewood, NJ, USA) tube immersed in deionized water inside a closed glass container.

All the tests in this study were conducted at 23.2 ± 0.349 °C and 24.3 ± 0.489% RH. To obtain relatively stable formaldehyde concentration and sensor signals, we implemented 75 min as the sampling time for the SFA30 sensors and 30 min for the Gravity sensors. This stabilization time was based on our experience with the sensors. The sensor signals from the Arduino interface were recorded and post-processed by taking the 15 min average of each. We also used the corresponding averaging time for the BBCEAS measurements.

### 2.2. Interference Gas Selection

Our previous study [10] of the SFA30 sensors performed preliminary cross-sensitivity tests for common trace gases, which included carbon monoxide (CO, 39.7 ppm), isobutylene (100 ppb), nitric oxide (NO, 101 ppb), and nitrogen dioxide (NO_2_, 83 ppb). In this study, we added ozone (O_3_) to the cross-sensitivity evaluation because of its local relevance. The target trace gas concentrations were determined using the 95th percentile measurements from the Utah Division of Air Quality (UDAQ) monitoring sites Hawthorne and Bountiful in the past two years (the percentile O_3_ concentrations were from the summer season). Table 3 shows the target trace gas concentrations evaluated in this study.

In addition to these common outdoor trace gases, we also considered methanol and IPA because these two gases can be present in indoor environments. Specifically, they are potentially present in anatomy labs that use embalming fluids. We measured the concentrations of formaldehyde, methanol, and IPA in an anatomy laboratory on the University of Utah campus over the course of 3 to 5 h (Appendix A) to obtain the target concentration range for the indoor VOCs. Specifically, formaldehyde concentrations were measured using Radiello^®^ (Padova, Italy) passive samplers [44,45], analyzed by the Desert Research Institute (DRI) with high-performance liquid chromatography (HPLC). We determined the concentrations of methanol and IPA using Assay Technology^®^ 545 Methanol Badges (Livermore, CA, USA) and 566 Organic Vapor Monitors (Livermore, CA, USA), respectively. These samplers were analyzed by Assay Technology with a gas chromatography flame ionization detector (GC/FID). Appendix A shows details of the passive sampling, including the concentrations of formaldehyde, methanol, and IPA in the anatomy lab. Based on these samples, we set the target formaldehyde concentration to 250 ppb; the target concentrations of methanol to 1, 2, or 3 ppm; and the target concentrations of IPA to 0.5, 1, or 1.5 ppm.

### 2.3. Methods for Concentration-Only Tests

We conducted concentration-only tests to evaluate sensor performance at target formaldehyde concentrations (23.2–326 ppb for the Sensirion sensors and 11.3–320 ppb for the Gravity sensors). The concentration ranges differed slightly for the two types of sensors because the BBCEAS measurements required post-processing at the end of each test to obtain formaldehyde concentrations.

All the sensors were evaluated both individually and as a group. A linear regression (LR) model provided the relationship between the sensor response and the reference measurements (BBCEAS):(1)$$\hat{y}=kc+b$$
where c is the formaldehyde concentration from the BBCEAS, k is the slope of the linear regression, b is the intercept of the linear regression, and $\hat{y}$ is the predicted sensor response based on the LR model.

We also investigated the sensor performance metrics [10] of the two low-cost sensors, which included the linearity of response, LOD, sensor accuracy, and sensor precision.

a. Linearity of response

In this study, we used the coefficient of determination (R^2^) to evaluate the linearity of response:(2)$$R^{2}=1-\frac{\sum {\left(y_{i}-{\hat{y}}_{i}\right)}^{2}}{\sum {\left(y_{i}-\bar{y}\right)}^{2}}$$
where $y_{i}$ is the observed sensor signal, $\bar{y}$ is the mean of the observed sensor signal, and ${\hat{y}}_{i}$ is the predicted sensor signal.

b. Limit of detection

The LOD was calculated based on the linear relationship between each sensor and the BBCEAS at low formaldehyde levels (11.3–60.8 ppb):(3)$$LOD=\frac{3.3\sigma }{k}$$
where σ is the standard error of the linear regression and k is the slope of the linear regression from Equation (1).

c. Sensor accuracy

We implemented the root mean squared error (RMSE) and the normalized root mean squared error (NRMSE) to evaluate sensor accuracy:(4)$$RMSE=\sqrt{\frac{1}{N}\sum_{i=1}^{N}{\left(c_{i}-{\hat{x}}_{i}\right)}^{2}}$$(5)$$NRMSE=\frac{RMSE}{c_{max}-c_{min}}$$
where $c_{i}$ is the formaldehyde concentration from the BBCEAS, ${\hat{x}}_{i}$ is the predicted formaldehyde concentration based on the sensor signal, $c_{max}$ is the maximum of the BBCEAS measurement, $c_{min}$ is the minimum of the BBCEAS measurement, and N is the total number of measurements.

d. Sensor precision

The coefficient of variation (CV) is a common metric for low-cost sensor precision. We repeated the concentration-only tests three times, and CV is given by the following equation:(6)$$CV=\frac{\sigma }{\mu }\times 100\%$$
where σ is the standard deviation of repeated measurements and μ is the mean of repeated measurements.

In this study, we could not reproduce the target formaldehyde concentrations. First, the dilution rate of the permeation oven could not be precisely controlled. Second, the formaldehyde concentration could not be dynamically adjusted because the BBCEAS provided formaldehyde concentrations only after post-processing at the end of each test. Consequently, we conservatively estimated the CV by calculating the means and standard deviations of the sensor responses for each target formaldehyde concentration of the three repeated tests.

### 2.4. Methods for Cross-Sensitivity Evaluation

Table 4 shows the reference/research-grade instrument for each interference gas.

We calculated a normalized sensor signal (NSS) to allow for comparing sensor performance when we could not exactly match the target formaldehyde concentrations.(7)$$NSS=\frac{sensor\;signal}{BBCEAS}$$

Consequently, the cross-sensitivity (CS) of the sensor to each gas was given by the following [46]:(8)$$CS=\frac{{NSS}_{after}-{NSS}_{before}}{{NSS}_{before}}\times 100\%$$
where ${NSS}_{after}$ is the normalized sensor signal after mixing the interference gas and ${NSS}_{before}$ is the normalized sensor signal before mixing the interference gas.

## 3. Results and Discussion

### 3.1. Concentration-Only Tests

Figure 2 and Appendix A show the linear the response of each individual SFA30 and Gravity sensor to formaldehyde concentrations ranging from 11.3 to 326 ppb. We also evaluated each type of sensor as a group. Appendix A shows the individual sensor metrics, and Table 5 shows the group metrics. As a group, the SFA30 and the Gravity sensors exhibited slopes of 13.2 ± 0.356 and 0.462 ± 0.0141, respectively. Although no criteria for low-cost formaldehyde sensors have been reported, the EPA recommends a slope of 1.0 ± 0.35 and an intercept range from −5 to 5 µg/m^3^ for field applications of the low-cost particulate matter sensors [47]. Hu et al. reported a slope of 1.04 for their LaFeO_3_ thin-film formaldehyde sensor over the formaldehyde concentration range of 0.05–1 ppm [29]. Yang et al. reported a slope of 1.37 for their formaldehyde sensors based on Ni-doped SnO_2_ nanoparticles over the formaldehyde concentration range of 0–50 ppm [48]. Crispi and Neri reported a slope of 0.248 for their Al, Ca-doped ZnO sensor over the formaldehyde concentration range of 0.125–4 ppm [31]. The slopes and the intercepts of both sensors evaluated in this study indicate non-ideal laboratory performance, which may lead to elevated sensor bias in field applications.

a. Linearity of response

As shown in Table 5 and Appendix A, both sensors exhibited good linearity as a group (R^2^ = 0.936 for the SFA30 sensors and R^2^ = 0.959 for the Gravity sensors). Individual sensors exhibited an R^2^ > 0.97 for the SFA30 sensors and an R^2^ > 0.96 for the Gravity sensors.

Previous laboratory evaluation studies have reported linearity of response by comparing sensor response to reference instruments using linear regression models. Hu et al. reported an R^2^ = 0.993 for their LaFeO_3_ film sensor over the formaldehyde concentration range of 0.05–1 ppm [29]. Crispi and Neri reported an R^2^ = 0.999 for their Al, Ca-doped ZnO sensor over the formaldehyde concentration range of 0.125–4 ppm [31]. An R^2^ value exceeding 0.75 is generally considered a strong agreement between a low-cost sensor and a reference instrument [21,23].

b. Limit of detection

The LOD was calculated based on the linear regression of the sensor response vs. the BBCEAS at low formaldehyde levels (11.3–60.8 ppb). As shown in Appendix A and Table 5, the individual LOD of the SFA30 and the Gravity sensors ranged from 4.58 to 6.58 ppb and from 4.73 to 14.5 ppb. As a group, the sensors exhibited an LOD of 13.7 ppb and 21.9 ppb for the SFA30 and the Gravity sensors, respectively.

The sensors evaluated in this study exhibited LODs that were lower than those reported in previous laboratory studies. Alolaywi et al. reported a formaldehyde LOD of 60 ppb for their electrochemical sensor [49]. Crispi and Neri reported an LOD of 125 ppb for their conductometric sensor [31]. A few other studies on nanomaterial-based sensors reported LODs ranging from 50 to 90 ppb [30,36,48]. The LODs in these previous studies tended to be significantly higher than typical indoor or outdoor levels. As an example, Hun et al. reported an indoor formaldehyde concentration of 17.2 ± 4.49 ppb from an evaluation of 179 residences in the U.S. [24]. Typical outdoor formaldehyde concentrations in urban areas of the U.S. range from 11 to 20 ppb [50], with some industrial areas of the U.S. reporting higher levels of formaldehyde (24–66 ppb) [50,51]. The SFA30 sensors exhibit lower LODs than the Gravity sensors, potentially making them more suitable for concentration ranges that are typical for indoor and outdoor urban areas.

c. Sensor accuracy

We used RMSE and NRMSE to characterize sensor accuracy (Appendix A and Table 5). The individual RMSE of the SFA30 and the Gravity sensors ranged from 14.9 to 16.1 ppb (NMRSE: 4.93–5.32%) and from 16.1 to 17.5 ppb (NRMSE: 5.20–5.65%), respectively. The group RMSEs of these two sensors were 24.1 ppb (NRMSE: 7.95%) and 19.1 ppb (NRMSE: 6.18%), respectively. Because only two Gravity sensors could be evaluated, the group RMSEs and NRMSEs may be greater than reported here.

Limited laboratory evaluations have reported RMSEs or NRMSEs for low-cost formaldehyde sensors. Song et al. reported an RMSE of 892 ppb for their yttrium-doped ZnO sensor array over the formaldehyde concentration range of 5–45 ppm. Although no criteria for low-cost formaldehyde sensors have been reported, for comparison purposes, the EPA’s criteria are an RMSE ≤ 5 ppb for low-cost ozone sensors [52] and an NRMSE ≤ 30% for particulate matter sensors [47]. Both the SFA30 and the Gravity sensors exhibited lower NRMSEs than the EPA’s criteria. Our results suggest that both sensors can provide reasonably accurate measurements if corrected using the slopes and intercepts shown in Appendix A.

d. Sensor precision

As discussed in Section 2.3, we conservatively estimated the CVs of the low-cost sensors (Table 5 and Appendix A). The SFA30 and the Gravity sensors exhibited group CVs of 4.50% and 1.13%, respectively. Individual CVs of the SFA30 and the Gravity sensors ranged from 3.83 to 5.46% and from 0.714 to 1.55%, respectively. Although no criteria for low-cost formaldehyde sensors have been reported, for discussion purposes, we compared the CVs of the two sensors to the EPA’s criteria (CV ≤ 30%) for both ozone and particulate matter sensors [47,52]. Our conservative estimations indicate that both sensors have acceptable precision at the formaldehyde levels evaluated in this study.

### 3.2. Cross-Sensitivity Evaluation

As discussed in Section 2.2 and Section 2.4, we determined the cross-sensitivity to outdoor trace gases and selected indoor VOCs by calculating the change in the normalized sensor signals after mixing the interference gases.

a. Outdoor trace gases

Figure 3 and Appendix A show the cross-sensitivity of the SFA30 and the Gravity sensors to outdoor trace gases at low formaldehyde levels (17.1–27.2 ppb). Among the five trace gases, NO_2_ (38.5 ± 0.891 ppb) and O_3_ (68.6 ± 0.877 ppb) significantly reduced the SFA30 sensors signals, resulting in an average cross-sensitivity greater than −98% for both gases. The SFA30 sensors exhibited an average cross-sensitivity of −36.4% to NO (33.4 ± 0.587 ppb) and limited cross-sensitivity (less than 1%) to CO (547 ± 129 ppb) and isobutylene (108 ± 4.06 ppb). These limited cross-sensitivities were lower than the variations in the BBCEAS signals (7.41% and 8.09%, respectively) during the experiments, and these were minor compared to the cross-sensitivities to NO, NO_2_, and O_3_. The Gravity sensors, however, exhibited low cross-sensitivity (less than 6% on average) to the outdoor trace gases. In fact, the average cross-sensitivity to each trace gas was lower than the BBCEAS variations (Appendix A).

Few studies have reported the sensor cross-sensitivity of low-cost formaldehyde sensors to common trace gases. Crispi and Neri reported that their Al, Ca-doped ZnO sensor exhibited about twice the response to NO_2_ (5 ppm), about one-fifth of the response to NO (125 ppm), and a negligible response to CO (25 ppm) compared to formaldehyde (4 ppm) [31]. Deng et al. found that their SnO_2_ sensor showed almost 1/10,000 of the response to NO_2_ compared to formaldehyde (both gas concentrations: 10 ppm at 120 °C) [35]. These studies investigated cross-sensitivity by comparing sensor response to formaldehyde and to another interference gas alone. They did not test the sensor response to the mixture of formaldehyde and the interference gas. Additionally, these previous studies did not include O_3_, which is an important atmospheric trace gas.

Previous studies have also reported the cross-sensitivity of other types of electrochemical sensors to common air pollutants. Mead et al. reported the cross-sensitivity of NO_2_ sensors to O_3_ (100%) [53]. Lewis et al. reported the cross-sensitivity of CO sensors to O_3_/NO_2_, NO sensors to O_3_/NO_2_, and O_3_ sensors to SO_2_/NO/NO_2_ in their laboratory evaluation [54]. Since the SFA30 and the Gravity sensors in this study are both electrochemical sensors, it is important to understand their cross-sensitivity to common outdoor trace gases. Our results show that the SFA30 sensors exhibited severe interference to NO_2_ and O_3_ and some interference to NO. The Gravity sensors exhibited limited cross-sensitivity to the common outdoor trace gases.

b. Indoor VOCs

We investigated the sensor cross-sensitivities to two target indoor VOCs at high formaldehyde levels (249–263 ppb). During this portion of the study, we found that the SFA30 sensors exhibited baseline drift at high formaldehyde levels throughout the 75 min test period. Specifically, the sensor signals first reached a peak and then continued to decrease throughout the test period (Figure 4). To address this drift, we implemented a formaldehyde baseline correction by running three tests of the same duration with only formaldehyde and fitting a linear regression model to the results. Consequently, we set the average sensor response (to formaldehyde only) as the baseline and compared it to the sensor response in the presence of interference gases. It should be noted that we found no baseline drift for the Gravity sensor measurements.

Figure 5 and Appendix A show the cross-sensitivities of the SFA30 sensors and the Gravity sensors to the two indoor VOCs (methanol and IPA). The SFA30 sensors exhibited less than 5% cross-sensitivity to methanol and IPA. However, the Gravity sensors exhibited greater cross-sensitivity to the two VOCs, greater than 118% to methanol and greater than 18% to IPA, respectively.

Previous studies have reported sensor cross-sensitivity to some VOCs. Hu et al. reported that their LaFeO_3_ thin-film sensors exhibited about one-eighth of the response to methanol compared to formaldehyde (both gases at 100 ppm) [29]. Alolaywi et al. reported that their electrochemical sensor exhibited about 1/100 of the response to methanol and IPA compared to formaldehyde (all gases at 0.1 ppm) [49]. Deng et al. reported that their CuO nanomaterial-based sensor exhibited almost twice the response to methanol (5 ppm) compared to formaldehyde (100 ppb) at the working temperature of 250 °C [30]. Li et al. reported that their TiO_2_ sensors exhibited about one-fifth of the response to methanol compared to formaldehyde (both gases at 1 ppm) [34]. Gautam et al. reported that their nanocomposite-based sensor exhibited about half the response to methanol compared to formaldehyde (both gases at 1 ppm) [28]. Li et al. also reported that their SnO_2_ sensor exhibited about half the response to methanol compared to formaldehyde (both gases at 100 ppm, 200 °C) [33]. These studies indicated that methanol often showed interference to low-cost formaldehyde sensors when evaluating sensor cross-sensitivity to indoor VOCs.

The SFA30 and the Gravity sensors exhibited different cross-sensitivities to outdoor trace gases and indoor VOCs. Specifically, the SFA30 sensors exhibited higher cross-sensitivities to NO, NO_2_, and O_3_, while the Gravity sensors exhibited higher cross-sensitivities to methanol and IPA. The differences in sensor cross-sensitivities may result from design differences in the electrochemical sensors, including design and material choices for electrodes, membranes, and electrolytes [55,56,57,58,59]. Although the two sensors were both electrochemical sensors, specific information about their design and material choices was not available. In addition, we also observed sensor-to-sensor cross-sensitivity variability (Appendix A) that was in the same range as the variation in formaldehyde concentration indicated by the reference monitor (7.16–10.7%). Mijling et al. reported significant sensor-to-sensor variability among NO_2_ sensors, even if the sensors were from the same manufacturing batch [60]. Overall, the cross-sensitivity of the Gravity sensors may limit their use for formaldehyde measurements in the presence of methanol. The SFA30 sensors exhibited some potential for indoor formaldehyde measurements in locations with methanol and IPA but may require field evaluation to understand the sensor performance (including sensor drift) at high formaldehyde levels.

### 3.3. Limitations

This study has several limitations. First, we evaluated a different batch of SFA30 sensors than the previous paper [10], which demonstrated different sensor performance. The SFA30 sensors evaluated in this study exhibited different slopes (13.2 ± 0.356) and intercepts (−369 ± 52.9) than those in the previous evaluation (1.08 ± 0.0188 and −6.94 ± 0.800, respectively). The group evaluation results of the Gravity sensors may have greater uncertainties because we were only able to test two Gravity sensors, compared to four SFA30 sensors. Second, we estimated the concentrations of methanol and IPA based on the flow rates from the MCFs rather than research-grade/reference instruments to measure the concentration of each gas because we did not have an instrument capable of measuring methanol and IPA at the concentrations. Furthermore, the cross-sensitivity tests only included a limited number of species (CO, NO, NO_2_, O_3_, isobutylene, methanol, and IPA) and a limited range of temperatures and RHs. Lastly, a long-term field evaluation is needed to understand sensor drift. However, this study helps by providing an improved understanding of sensor performance (including sensor drift) at a variety of formaldehyde levels and in the presence of possible interference gases.

## 4. Conclusions

This study performed a laboratory evaluation of four SFA30 sensors and two Gravity sensors, with the BBCEAS providing reference formaldehyde measurements. The sensors’ performance was evaluated both individually and as a group. In general, both sensors exhibited a good linearity of response (R^2^ > 0.93 as a group for both sensors) when they were exposed to formaldehyde alone. The SFA30 sensors and the Gravity sensors exhibited LODs of 13.7 ppb and 21.9 ppb as a group, respectively. Both sensors exhibited relatively good accuracy (group NRMSE < 8%) and good precision (group CV < 5%).

The SFA30 sensors exhibited severe cross-sensitivity to NO_2_ and O_3_ (greater than −98%) and some cross-sensitivity to NO (−36.4% in average), which would likely preclude the use of these sensors for ambient outdoor formaldehyde measurements. Although the Gravity sensors exhibited limited cross-sensitivity to the five outdoor trace gases, the LOD (21.9 ppb as a group) may be insufficient for ambient outdoor formaldehyde measurements. On the other hand, the Gravity sensors exhibited severe cross-sensitivity to methanol (greater than 118%). The SFA30 sensors exhibited limited cross-sensitivity (less than 5%) to methanol and IPA, although the sensors exhibited baseline drift and high sensor bias at high formaldehyde levels. The results suggest that the SFA30 sensor is unsuitable for outdoor formaldehyde measurements and that the Gravity sensor might be suitable for ambient locations with concentrations greater than typical ambient levels. However, the SFA30 sensor may be a useful screening tool for indoor environments, although a field evaluation is needed to understand the sensor’s performance (including sensor drift). The findings can help guide sensor deployment strategies depending on the environment, specifically indoor or outdoor locations. Lastly, combining the two types of sensors could be a promising strategy for some formaldehyde measurements.

## Figures and Tables

**Figure 1 sensors-25-03096-f001:**
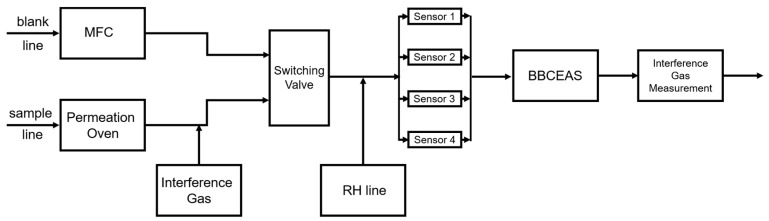
Flow chart of the laboratory evaluation system. All the tubing inside the system was PTFE tubing (OD ¼″), and all fittings were stainless steel.

**Figure 2 sensors-25-03096-f002:**
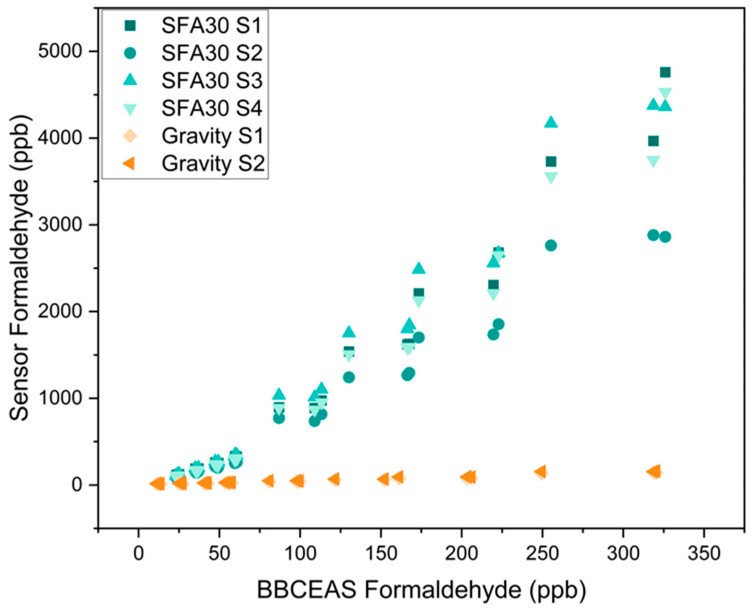
Response of individual SFA30 and Gravity sensors to formaldehyde in the three repeated concentration-only tests. Each response shown in the figure is the average of 15 min stable sensor signals and BBCEAS measurements (Appendix A). All tests were performed at 23.2 ± 0.349 °C and 24.3 ± 0.489% RH.

**Figure 3 sensors-25-03096-f003:**
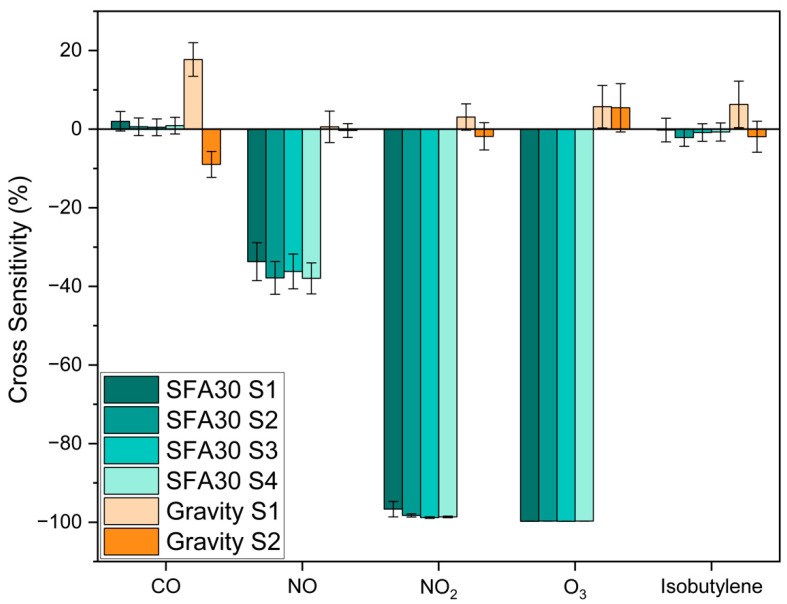
Cross-sensitivity of the SFA30 and the Gravity sensors to outdoor trace gases at low formaldehyde levels (17.1–27.2 ppb), 23.2 ± 0.349 °C and 24.3 ± 0.489% RH. Error bars denote the standard deviations of three repeated tests for each gas.

**Figure 4 sensors-25-03096-f004:**
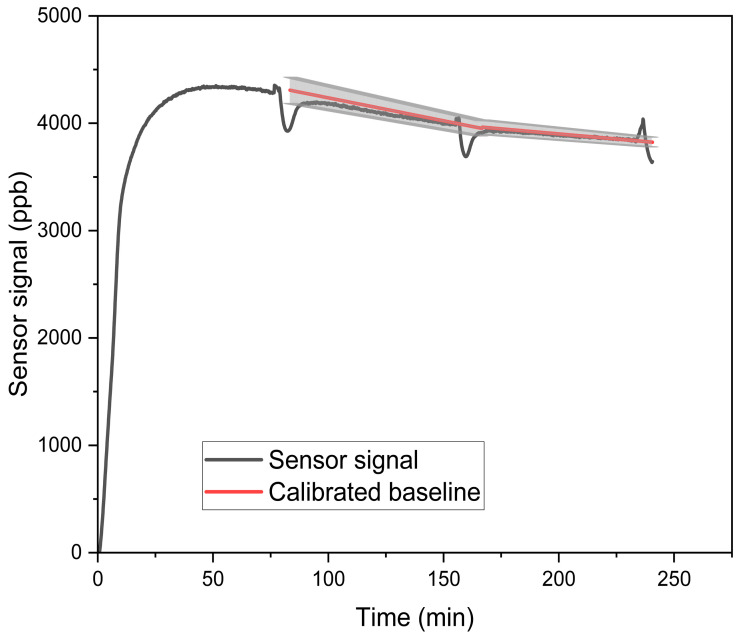
Baseline correction at high formaldehyde levels (249–263 ppb) for tests of the SFA30 sensors. Solid black line denotes the uncorrected sensor signal. Shaded area denotes the standard deviation of the calibrated baseline (average of 3 runs).

**Figure 5 sensors-25-03096-f005:**
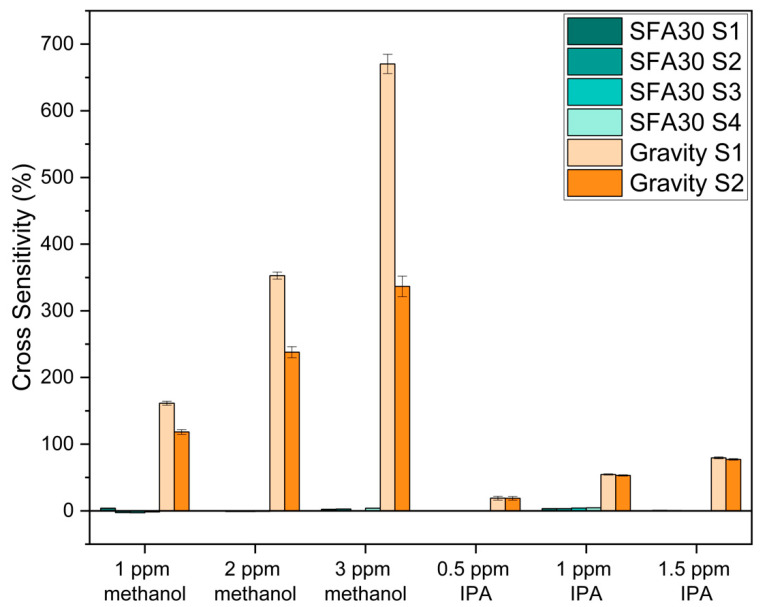
Cross-sensitivity of the SFA30 and the Gravity sensors to methanol and IPA at high formaldehyde levels (249–263 ppb), 23.2 ± 0.349 °C and 24.3 ± 0.489% RH. Error bars denote the standard deviations of the 15 min cross-sensitivity calculations for each test.

**Table 1 sensors-25-03096-t001:** Laboratory evaluation of low-cost formaldehyde sensors.

Reference	Sensor Type	Concentration (ppb)	Temperature (°C)	RH (%)
Pei et al. [10]	EC	0–76	0–40	0–75
Alonso et al. [26]	Micro fuel cell	17.3–477	19.9–27.7	26.8–72.6
Chattopadhyay et al. [27]	EC/MO	0–800	22–50	8–85
Gautam et al. [28]	MO	10–1,000,000	Room temperature	49 ± 5
Hu et al. [29]	MO	50–1000	90–150	N/A
Deng et al. [30]	MO	50–1000	160–320	25
Crispi and Neri [31]	MO	125–4000	100–400	N/A
Descamps et al. [32]	Fluorescence-based	0–192	15–35	0–72.5
Li et al. [33]	MO	1000–500,000	200	25–30
Li et al. [34]	MO	124–5000	Room temperature	40–50
Deng et al. [35]	MO	1000–100,000	30–200	20–90
Song et al. [36]	MO	80–5000	140–180	11–75

Note: EC denotes electrochemical; MO denotes metal oxide; N/A denotes not available.

**Table 2 sensors-25-03096-t002:** Sensor specifications.

Sensor	SFA30	Gravity	PS1-HCHO-1
Manufacturer	Sensirion AG	DFRobot	SGX Sensortech
Sensor type	Electrochemical	Electrochemical	Electrochemical
Price	USD 45	USD 50	USD 150
LOD *	<20 ppb	Not reported	10 ppb
Publications that used/evaluated the sensor	Pei et al. [10] El-Leathey et al. [38] Crispi and Neri [31] Fissore et al. [39]	Bapat et al. [40] Marques and Pitarma [41] Corona et al. [42] Fu et al. [43]	Not reported

* LOD from the manufacturer’s datasheet.

**Table 3 sensors-25-03096-t003:** Common trace gas concentrations for the cross-sensitivity evaluation.

	Gas	Concentration *
Outdoor trace gas (ppb)	CO	547 ± 129
	Isobutylene	108 ± 4.06
	NO	33.4 ± 0.587
	NO_2_	38.5 ± 0.891
	O_3_	68.6 ± 0.877
Indoor VOCs (ppm)	Methanol	1, 2, or 3
	IPA	0.5, 1, or 1.5

* The concentrations are presented as ± the standard deviation of each target gas concentration as measured by the reference/research-grade instruments when available.

**Table 4 sensors-25-03096-t004:** Reference/research-grade instruments for the cross-sensitivity evaluation.

Target Gas	Reference/Research-Grade Instruments
CO	Q-Trak 7575, TSI, Shoreview, MN, USA
Isobutylene	
NO	Model 42i NO-NO_2_-NO_x_ Analyzer, Thermo Fisher Scientific, Waltham, MA, USA
NO_2_	
O_3_	Model 106-L ozone monitor, 2B Technologies

**Table 5 sensors-25-03096-t005:** Group performance metrics for the concentration-only tests.

Metrics	SFA30	Gravity
Slope *	13.2 ± 0.356	0.462 ± 0.0141
Intercept *	−369 ± 52.9	3.86 ± 1.99
R^2^	0.936	0.959
RMSE (ppb)	24.1	19.1
NRMSE (%)	7.95	6.18
CV (%)	4.50	1.13
LOD (ppb)	13.7	21.9

* The slopes and intercepts denote k and b of Equation (1), respectively (Section 2.3).

## Data Availability

The data presented in this study may be obtained from the authors upon reasonable request.

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
