# Peer review of "Laboratory Cross-Sensitivity Evaluation of Low-Cost Electrochemical Formaldehyde Sensors"

_sensors, 2025, doi:10.3390/s25103096_

Round 1

Reviewer 1 Report

Comments and Suggestions for Authors

the work is well written and well organized . please add the following

  1. why those sensors show changes in cross sensitivty ?
  2. mechanism of detection affects the cross ensitivity
  3. is there any relation between amnufacturing process and cross sensitivity ?
Comments on the Quality of English Language

typo erros especially in fiugure and tbale numbers should be revised.

Reviewer 2 Report

Comments and Suggestions for Authors

This research article provides a comprehensive study for laboratory cross-sensitivity evaluation of low-cost electrochemical formaldehyde sensors. The authors evaluated the sensors in a mixture of formaldehyde with each of five outdoor trace gases (CO/NO/NO2/O3/isobutylene) and two indoor VOCs (methanol and isopropyl alcohol). The results suggest that the Gravity sensors may be useful for outdoor formaldehyde measurements when formaldehyde levels are well above background levels, and the SFA30 sensors may be a useful screening tool for indoor environments. This paper could be published in Sensors after minor revision.

  1. Clearly explain whether calibration curves were generated once or recalibrated during/after stabilization or drift correction phases.
  2. Provide recommendations for field users regarding how to address baseline drift observed in the SFA30 sensor (e.g., scheduled recalibration, baseline monitoring protocols).
  3. Add a schematic or photo of the custom-made 3D-printed sensor chamber to help readers visualize the experimental setup.
  4. Add an extra sentence emphasizing that the findings can guide deployment strategies depending on the environment (indoor vs outdoor, VOC background).
  5. Please provide references for Eq. 7 and 8

Doi: 10.1021/acssensors.4c02097

  1. Carefully proofread for minor typos and inconsistencies (e.g., "TheEl-Leay" should be corrected to "El-Leathey" in references).

Reviewer 3 Report

Comments and Suggestions for Authors

The manuscript studies the sensitivity of industrially produced electrochemical sensors to formaldehyde and other gases in both indoor and outdoor ambient atmospheres. The importance of this work lies in the negative impact formaldehyde has on human health and the need for its concentration control.

Through the measurements conducted by the authors, features and limitations of each type of sensor were revealed. Specifically, SFA30 sensor signals are affected by oxidizing gases like ozone and nitrogen dioxide, while Gravity sensors show stability to these gases but are susceptible to methanol. Based on these findings, recommendations for using each type of sensor are provided. Additionally, all the obtained values for the characteristics of formaldehyde detection, linearity of response, sensor accuracy, limit of detection, and sensor precision were compared to the results of other studies, and relevant links are provided. The conclusions are based on experimental data.

To further enhance the understanding of this work, the authors should make a few additions:

1. Provide the schemes of the studied sensors.

2. According to the measurement methodology, the authors did not specify which sensor signal was used.

3. Make assumptions about the reasons for the different cross-sensitivity of the studied sensors.

In the future, authors should consider the effect of humidity on the signals from the sensors they are studying.
